# Discrimination and Prediction of *Lonicerae japonicae Flos* and *Lonicerae Flos* and Their Related Prescriptions by Attenuated Total Reflectance Fourier Transform Infrared Spectroscopy Combined with Multivariate Statistical Analysis

**DOI:** 10.3390/molecules27144640

**Published:** 2022-07-20

**Authors:** Yang-Qiannan Tang, Li Li, Tian-Feng Lin, Li-Mei Lin, Ya-Mei Li, Bo-Hou Xia

**Affiliations:** Key Laboratory for Quality Evaluation of Bulk Herbs of Hunan Province, Hunan University of Chinese Medicine, Changsha 410208, China; tangyangqiannan@163.com (Y.-Q.T.); licorece@hotmail.com (L.L.); m13460513070@163.com (T.-F.L.); lizasmile@163.com (L.-M.L.); yameili@hnucm.edu.cn (Y.-M.L.)

**Keywords:** *Lonicerae japonicae Flos*, *Lonicerae Flos*, ATR-FTIR, multivariate statistical analysis

## Abstract

LJF and LF are commonly used in Chinese patent drugs. In the *Chinese Pharmacopoeia*, LJF and LF once belonged to the same source. However, since 2005, the two species have been listed separately. Therefore, they are often misused, and medicinal materials are indiscriminately put in their related prescriptions in China. In this work, firstly, we established a model for discriminating LJF and LF using ATR-FTIR combined with multivariate statistical analysis. The spectra data were further preprocessed and combined with spectral filter transformations and normalization methods. These pretreated data were used to establish pattern recognition models with PLS-DA, RF, and SVM. Results demonstrated that the RF model was the optimal model, and the overall classification accuracy for LJF and LF samples reached 98.86%. Then, the established model was applied in the discrimination of their related prescriptions. Interestingly, the results show good accuracy and applicability. The RF model for discriminating the related prescriptions containing LJF or LF had an accuracy of 100%. Our results suggest that this method is a rapid and effective tool for the successful discrimination of LJF and LF and their related prescriptions.

## 1. Introduction

LJF, widely used in common Chinese medicine, is the flower bud of *Lonicera japonica* Thunb. mainly produced in Shandong, Henan, and Hebei Provinces in China [1]. It is commonly applied in the treatments of sores, furuncles, carbuncles, swelling, and infections caused by exopathogenic wind-heat or epidemic febrile diseases [1]. At the same time, there is another Chinese medicine named LF, which is defined by the *Chinese Pharmacopeia* as the dried flower bud of *Lonicera macranthoides* Hand.-Mazz., *Lonicera fulvotomentosa* Hsu et S. C. Cheng, *Lonicera hypoglauca* Miq., and *Lonicera confusa* DC. These four LF species are mainly grown in the south of the Yangtze River, including the Hunan, Guizhou, Guangxi, and Guangdong Provinces in China [1]. LF possesses a significant antipyretic effect, improves liver functions, and exhibits an antibacterial effect [2]. Although LJF and LF are widely used herbs with similar phenotypes derived from the plants of the same genus, LJF and LF are listed as independent items in the *Chinese Pharmacopeia* (2020 Edition) for medical safety, especially regarding drug injection. *Chinese Pharmacopeia* (2005 Edition) lists LJF and LF as independent items because LF and LJF are significantly different in medicinal history, plant morphology, medicinal properties, and chemical constituent, and the only plant source of LJF is again limited to *Lonicera japonica* Thunb. In contrast, LF has three plant sources, including *Lonicera macranthoides* Hand.-Mazz., *Lonicera hypoglauca* Miq., and *Lonicera confusa* DC. Moreover, *Chinese Pharmacopeia* (2010 Edition) adds another plant source named *Lonicera fulvotomentosa* Hsu et S. C. Cheng to LF following the 2005 Edition, obtaining a total of four plants of the same genus under the legal species of LF.

According to traditional records, LJF and LF can clear heat and detoxify. Thus, they are often applied for the treatment of sores, furuncles, carbuncles, and issues caused by exopathogenic wind-heat or epidemic febrile diseases. Based on modern pharmacological studies on LJF and LF, there are some different medicinal effects between LJF and LF. Compared with other commonly seen antibacterial drugs, LJF manifests more powerfully in antibacterial activity and inhibition of drug-resistant bacteria [2]. LJF has glucose-lowering, anti-ultraviolet radiation, anti-endotoxin, anti-ulcer, anti-early pregnancy, anti-platelet aggregation, anti-fertility, and neuroprotective activities that are not reported in LF, while LF has effects on balancing intestinal flora and anti-atherosclerotic effects that are not reported in LJF [3]. In recent years, the commercial value of LJF in herbal medicine trading markets has increased by over 400%, and more than 30% of current traditional Chinese medicine prescriptions contain LJF [4]. As a result, LF is often misused as LJF by some pharmaceutical factories to obtain higher interest. Furthermore, due to the difference in pharmacological activity between LJF and LF, the related prescriptions of both have different pharmacological effects. The incorrect addition of LJF and LF to their related prescriptions could cause great damage to human health and industrial development. Therefore, it is essential to develop a rapid identification method to discriminate between LJF, LF, and their related prescriptions to ensure the safety of patients and the healthy development of *Lonicera*-based trade markets.

Previous studies on the identification of LJF and LF have mainly focused on morphology and anatomical characters [5], chemical analysis [4], and DNA molecular marker techniques [6]. According to research, liquid chromatography and mass spectrometry were used to identify *Lonicera* species flower buds [7,8]. Although these methods have some advantages in the identification of *Lonicera* origins, they are not suitable for online quality control and are time-consuming. The sample pretreatment and detection processes of LC-MS and NMR are complicated, and the absorption bands of FT-NIRs are relatively broad, and the absorption bands are seriously overlapped, which means they cannot be directly performed in qualitative analysis. Recently, the potential of the hyperspectral imaging method is applied for the rapid identification of true and false honeysuckle tea leaves [9]. They are mainly concerned with the study of LJF and LF as tea leaves. Tea belongs to the field of food. However, the LJF- and LF-related pharmaceutical industry is a big business in China. The adulteration problem in the pharmaceutical field is far more important than that in the food field, and the harm is great. Therefore, it is urgent to supplement and modify some research gaps in this field.

ATR-FTIR technology has been developed based on FTIR technology, which can be equipped with ATR accessories for crystal materials [10], and is a fast, convenient, non-destructive, and high-throughput analytical tool that is widely used in the rapid identification and quality control of herbal medicines [11]. However, the obtained FT-IR spectra can be affected by the characteristics of samples in thickness and particle size [12,13], so the acquired raw data should be normalized. Compared with other kinds of infrared spectroscopy, ATR-FTIR is more convenient, simple, and effective. The use of this high-performance ATR accessory does not need time-consuming sample preparation and specialized sample holders such as the KBr pellet method [7,14]. In addition, the detected samples stay in their natural state, whose spectrum could reflect the original chemical information of various metabolites. ATR-FTIR also has excellent sample-to-sample reproducibility and minimal operator-induced variations [8]. In recent years, ATR-FTIR spectra combined with multivariate statistical analysis have been widely applied for quality assessment and authentication of herbal medicines [15,16,17]. However, the original ATR-FTIR spectra contained some overlapped absorbance and extra absorbance caused by different particle sizes and thickness of sample powders, so it was necessary for them to be preprocessed by spectral filter transformations and normalization methods.

In this work, a comprehensive study of the differences between LJF, LF, and their related prescriptions was proposed based on ATR-FTIR. The spectral process methods (spectral filter transformations and normalization methods), different pattern recognition methods, model parameters, and variable selection methods were optimized to form the optimal distinguishing prediction model. Spectral filter transformations include MSC, SNV, Savitzky–Golay, Row Center, and EWMA. Three normalization methods are applied to ATR-FTIR spectral data: area normalization, min-max normalization, and vector normalization. To construct a more precise discrimination model, we selected an optimal pattern recognition method from RF, SVM, and PLS-DA.

## 2. Materials and Methods

### 2.1. Materials and Sample Preparation

LJF and LF dried bud samples recorded in the *Chinese Pharmacopeia* (2020 Edition) were collected from the main *Lonicera*-producing areas in Hunan (*Lonicera macranthoides* Hand.-Mazz.) and Shandong in China, with all samples identified by Prof. Li-Min Gong from the Hunan University of Chinese medicine. Chinese patent medicines containing LJF or LF were purchased from the authentication-passed pharmacies. The ATR-FTIR, LJF, and LF dried buds samples were dried at 60 °C [18] to constant weight (the weight variation was less than 0.1%) in an electric thermostatic drying oven to ensure that moisture was not an interfering factor. Samples were then ground into powders; each sample was finely powdered by agate mortar and screened with a 200-mesh stainless steel sieve. The Chinese patent medicines samples were also ground into powders and then screened with a 200-mesh stainless steel sieve. All samples were stored in a relatively dry environment. 

### 2.2. Spectral Acquisition and Data Preprocessing

Each powder was subjected to an FTIR spectrometer (Nicolet iS5, Thermo Scientific, Waltham, MA, USA) equipped with an ATR accessory for recording the FTIR spectrum. The OMNIC program (version 8.2.0.387, Thermo Scientific, Waltham, MA, USA) was used to obtain all ATR-FTIR spectra. In total, 64 scans were recorded to obtain average analytical results and improve the signal-to-noise ratio. The transmittance of each spectrum was collected between 4000 and 600 cm^–1^ with a spectral resolution of 4 cm^–1^. 

ATR-FTIR spectral filter transformations, including MSC, SNV, Savitzky–Golay, Row Center, and EWMA normalization, were applied to minimize the baseline noise and maximize the differences found in spectra. The normalization methods were area normalization, minimum-maximum normalization, and vector normalization, which could reduce the effect of the physical characteristics of samples (particle size and thickness). In vector normalization, all spectra were converted from transmittance to absorbance, and the ATR-FTIR absorbance spectra were converted into the first and second derivatives with Savitzky–Golay derivative and nine smoothing points in OMNIC software. For vector normalization, the Euclidean norm was performed to calculate the absorbance values to acquire the normalization values of the spectra. In area and min-max normalizations, all spectra were converted from transmittance to absorbance, and then ATR correction was conducted by using OMNIC. For area normalization, each absorbance value at a specific wave number was divided by the total (integrated) absorbance area of the spectrum. For min-max normalization, each absorbance value was divided by the difference between the highest and lowest absorbance values. In addition, for the baseline correction, the airPLS algorithm was used to remove the baseline for all the spectra. The airPLS was proved to be a mature and effective algorithm [19].

### 2.3. Chemometrics Methods

#### 2.3.1. Random Forest (RF)

RF was developed by Breiman in 2001 and has been widely used to resolve classification problems or regression issues as a non-parametric algorithm based on a learning strategy (called ensemble learning) [20]. The RF model consists of hundreds of trees, and each tree was grown through a bootstrap sample of the original data. In addition, each node was selected by each tree and corresponded to a random variable subset. Each tree could provide a classification result to decide the final accurate category. The operational process of the RF model could be separated into the following steps.

Firstly, a spectra dataset was divided into the calibration set (bootstrap samples) and the validation set (OOB samples) with the help of the KS algorithm by MATLAB 2017a (MathWorks, Natick, MA, USA). Then, 4/5 of all LJF and LF samples were included in the calibration set, which was applied to obtain the optimal classification trees. The validation set was used to evaluate the ability of the RF model at last. Secondly, the values of n_tree_ and the square root of the number of all variables m_try_ based on the calibration set were both selected to acquire the optimal n_tree_ and m_try_. The OOB error was the rate of misclassification over all out-of-bag samples. The best n_tree_ was acquired based on the lowest OOB error, which was beneficial for further prediction. Thirdly, a new importance variable matrix was formed after exerting the optimal n_tree_ and m_try_. Fourthly, the new matrix was inserted in the producer, the original spectra variables were rearranged according to variable importance, and the most important variables were selected by a lower 5-fold cross-validation error rate. Finally, the validation set was inserted in the random forest model established based on the calibration set for the final model prediction. The establishment of the final RF discrimination model was performed by using the optimized n_tree_ and m_try_ parameters. The first two steps were rerun to calculate the final classification accuracy. Steps two to five were running in MATLAB.

#### 2.3.2. Support Vector Machine (SVM)

SVM is an effective classification method proposed by Vapnik, which is based on SLT and the principle of SRM [21]. SVM, as one of the kernel-based pattern recognition methods, has been successfully applied in the classification of drug and nondrug problems [22].

SVM can obtain nonlinear and global solutions even with the high-dimensional input vector [23]. An optimal classifier generalization is acquired when it minimizes training error along with higher testing accuracy for unknown testing datasets. The training algorithm of SVM maximizes the margin between class boundary and the training data by removing some meaningless data from the training dataset. So, the resulting decision function only depends on the training data called support vectors. Therefore, SVM maximizes the boundary by minimizing the maximum loss and gives good accuracy [24]. Experiments are performed in MATLAB using LIBSVM.

A kernel is a key that determines the performance of the SVM. The radial basis function (RBF) kernel was selected as the kernel function because of the excellent classification performance shown in previous studies [25,26]. The training data consist of the input matrix x_i_ (i = 1, 2, …, *n*) and an output vector y_i_ (i = 1, 2, …, *n*), where +1 and −1 are used to stand for the two classes. The SVM constructed an optimized linear regression by mapping the input vector x to a high-dimensional feature space by a nonlinear mapping using a kernel function K(x_i_, y_j_). The RBF kernel is expressed as K(x_i_, y_j_) = exp(−γ‖x_i_, y_j_‖^2^), where γ is the width parameter of the RBF kernel function [27]. There are several methods to the SVM to deal with classification problems, including “One-Versus-Rest (OVR)” [28], “One-Versus-One (OVO)” [29], and DAGSVM [30]. In this paper, we chose the OVO strategy, which can scale well to a large number of classes.

#### 2.3.3. Partial Least Squares Discrimination Analysis (PLS-DA)

PLS-DA, a binary classification algorithm from 0 to 1, is an adaptation of the PLS regression algorithm to the problem of supervised clustering. It has been extensively used in the analysis of multivariate datasets between independent and dependent variables, which are expressed by X and Y, respectively [31]. For PLS-DA, outliers were first eliminated by PCA combined with Mahalanobis distance. PCA A PLS-DA model was created on the training dataset using the preprocessed data. Experiments are performed in MATLAB.

### 2.4. Data Analysis

RF, SVM, and PLS-DA models can obtain the vote matrices. The three models in MATLAB could calculate the values of TN, TP, FN, and FP, respectively. Sensitivity, SENS (Equation (1)), Specificity, SPEC (Equation (2)), Accuracy, ACC (Equation (3)), and Matthew’s correlation coefficient, MCC (Equation (4)) were the four parameters for each class, indicating the identification effects for different samples of the three models. These four parameters with higher values indicate better identification ability for each class. The MCC is a correlation coefficient applied to evaluate the performance of binary classifications. It ranges from ±1 to 0; +1 indicates a perfect identification while 0 shows the performance of a random classification. Only binary PLS-DA classification models were calculated by the MCC [32].
(1)$$\mathrm{SENS}=\frac{\mathrm{TP}}{\left(\mathrm{TP}+\mathrm{FN}\right)}$$
(2)$$\mathrm{SPEC}=\frac{\mathrm{TN}}{\left(\mathrm{TN}+\mathrm{FP}\right)}$$
(3)$$\mathrm{ACC}=\frac{\left(\mathrm{TN}+\mathrm{TP}\right)}{\left(\mathrm{TP}+\mathrm{TN}+\mathrm{FP}+\mathrm{FN}\right)}$$
(4)$$\mathrm{MCC}=\frac{\left(\mathrm{TP}\times \mathrm{TN}-\mathrm{FP}\times \mathrm{FN}\right)}{\sqrt{\left(\mathrm{TP}+\mathrm{FP}\right)\left(\mathrm{TP}+\mathrm{FN}\right)\left(\mathrm{TN}+\mathrm{FP}\right)\left(\mathrm{TN}+\mathrm{FN}\right)}}$$

## 3. Results and Discussion

### 3.1. Band Assignment Comparison between LJF and LF

Spectral data were obtained in the range of 4000–600 cm^–1^. The representative samples of LJF and LF spectra are shown in Figure 1. The spectra were analyzed, and several noticeable peaks were observed. Assignments of the main absorption peaks are also summarized in Table 1.

The tiny band between 4000 and 3500 cm^–1^ corresponded to water-vapor O-H stretching, and the band was attributed to the O-C-O stretching of carbon dioxide at 2442–2208 cm^–1^ [33]. The stretching band of O-H near 3350 cm^−1^ and the stretching bands of C-O in the region of 1200–950 cm^−1^ indicated the existence of saccharides in LJF and LF [34]. LJF and LF spectra showed different patterns in this region, which meant the saccharides were different in LJF and LF. The peaks at 2920 cm^−1^ and 2851 cm^−1^ are attributed to the asymmetrical and symmetrical stretching bands of CH_2_- [35], in conjunction with the stretching band of C=O near 1729 cm^−1^ [36], indicating the existence of lipids. LJF and LF are covered by cuticles [1,37], which are continuous lipid membranes including cutin, waxes, cutan, and polysaccharides [38,39]. The absorption peaks near 2920, 2851, and 1729 cm^−1^ were from cutin and waxes [40], which consisted of acids, alcohols, esters, alkanes, etc. The peaks near 1630 cm^−1^ were due to C-O and C-N protein stretching [41]. This is known as the amide I band and is the main amide band. The peaks at 1440 and 1374 cm^−1^ were attributed to organic acid OH vibrational modes [42]. The peaks at 1321 and 1259 cm^−1^ were due to C–O stretching vibrations. The peak near 1150 cm^−1^ was assigned to CO-O-C asymmetric stretching of cholesterol ester and C-O stretching of oligosaccharides and triacylglycerols [43,44]. One was at 1051 cm^–1^ due to C-O stretching of starch [45].

Moreover, LJF showed the amide II bands at 1545 cm^−1^ while LF showed the aromatic skeletal bands near 1528 cm^−1^. This indicated that LJF contains more proteins, but LF contains more aromatic compounds (phenolic acids, flavonoids, etc.) [46,47]. In addition, LJF showed a peak at 1400 cm^−1^ and a weak peak at 930 cm^−1^, which were both absent in LF. The peak at 1400 cm^−1^ could be assigned to the bending mode of O-C-H, while the peak at 930 cm^−1^ could correspond to the skeletal mode of saccharides [48]. These two peaks indicated the different saccharides in LJF and LF. Another discriminating peak was near 780 cm^−1^, which was present in LF but not in LJF. This difference may be caused by the high content of saponins in LF. In summary, LJF and LF can be distinguished by the four peaks near 1545, 1400, 930, and 780 cm^−1^.

### 3.2. Classification of LJF and LF

In this study, RF, SVM, and PLS-DA models were applied in the classification and prediction of LJF and LF samples in a MATLAB programming environment. Spectra datasets were divided into the calibration set and the validation set with the help of the KS algorithm. The calibration set was 4/5 of all LJF and LF samples. In the training process of these three models, 5-fold cross-validation was used to demonstrate the performance of our methods.

As for establishing the RF model, the first step was to select the number of trees for the optimal classification trees model by the training process. Before this step, the initial raw dataset and normalization datasets were calculated with 300 trees to choose which dataset was better to establish the RF model. SVM and PLS-DA models were also established based on the raw dataset and normalization datasets. As shown in Table 2, RF models showed the performance by applying suitable vector normalization after the first differentiation in the calibration set. The sensitivity, specificity, accuracy, and MCC by the RF method reached 0.9706, 1, 0.9844, and 0.9692, respectively, which outperformed the other pretreatment methods. In terms of accuracies, the RF model also outperformed PLS-DA and SVM models in terms of overall accuracies. When compared with classifiers, the highest accuracy of RF was 0.0071 higher than that of PLS-DA and 0.0071 higher than that of SVM, respectively. Moreover, the validation sets for the assessment of the three models’ performance are shown in Table 3, so it can be seen that the RF model with the first derivative vector normalization was the optimal prediction model for discriminating LJF and LF, having accuracy of 0.9744.

Furthermore, the number of trees, the wave-number regions, and the variable importance cutoff value were employed to acquire the best prediction RF model. The optimal number of trees was obtained according to the lowest OOB classification error value. As shown in Table 4, these trees with the lowest error were in the numbers 100, 200, 300, 500, 800, and 1000. The number of grown trees with the highest accuracy was 300. Then, 300 trees were chosen for branch nodes selection (the m_try_ value). Using these 300 trees, the result of nodes calculation was acquired in Table 4. In total, 86 branch nodes were selected after inputting the optimal number of trees with the lowest OOB error and the highest accuracy of 0.9886 in the calibration set. The results indicated that 300 trees and 86 branch nodes were used for further training and prediction of the model.

Since ATR-FTIR spectra may be influenced by environmental factors including water vapor and carbon dioxide, RF models established by utilizing different wave-number ranges were compared to identify the best prediction model. As presented in Table 5, the ATR-FTIR spectral region between 4000 and 600 cm^−1^ was the best prediction model for discriminating LJF and LF since it had the highest accuracy (=0.9844). Furthermore, the original spectra variable was rearranged according to cutoff values. As listed in Table 6, the RF model had the higher four parameters data with a VIP cutoff value of 0.01, having the highest prediction accuracy of 0.9886. Thus, we could conclude that the RF method was selected as an appropriate classifier in this study with 300 trees, 86 branch nodes, 4000–600 cm^−1^ wavenumber area, and the VIP cutoff value of 0.01 under the first derivative vector normalization.

### 3.3. Classification of LJF- and LF-Related Prescriptions

LJF- and LF-related prescriptions are mainly composed of LJF and LF, respectively. Under the guidance of TCM theory, varied compound prescriptions, containing different herbs, are formulated based on the principle of “Emperor, Minister, Adjuvant, Courier”, a metaphor suggesting that different herbs have different effects and that the TCM practitioner should prescribe them according to the specific status of a patient. Due to the difference in ingredients, compound prescriptions differ in curative effects and side effects. Therefore, we can apply the methods of classification and prediction of LJF and LF in identifying LJF- and LF-related prescriptions. As mentioned above, we believe that the optimal prediction RF model of discriminating LJF and LF can show the appropriate performance in the identification of LJF- and LF-related prescriptions. After calculating LJF and LF traditional medicine ATR-FTIR spectra datasets by the RF model, four evaluation parameters (SENS, SPEC, ACC, and MCC) were all 1, which suggested that the RF model was an effective classifier for prescriptions. Thus, the present results indicate that the established RF model is the optimal predictable one to discriminate LJF and LF and their related prescriptions.

## 4. Conclusions

In this study, LJF and LF and their related prescriptions were analyzed by ATR-FTIR spectra combined with multivariate classification methods. The chemical information differences between LJF and LF were revealed in the light of band assignments based on the ATR-FTIR spectra. This is the first study that established classification models for the identification of LJF and LF using various factors, including spectral filter transformations, normalization methods, VIP cutoff, and wave-number region. The feasible results indicated that ATR-FTIR combined with RF is a rapid, efficient, reliable, and stable online classification and prediction method for LJF and LF and their related prescriptions, which has wide adaptability and can be applied in the application of Chinese medicine.

## Figures and Tables

**Figure 1 molecules-27-04640-f001:**
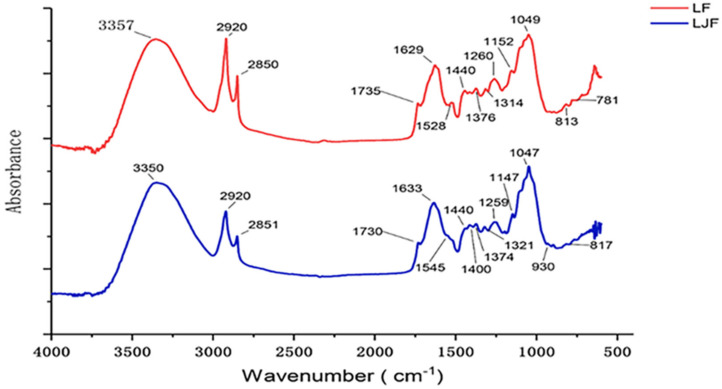
ATR-FTIR spectra of representive LJF and LF.

**Table 1 molecules-27-04640-t001:** Peak assignments of the ATR-FTIR spectra of LJF and LF.

LJF (cm^−1^)	LF (cm^−1^)	Vibration	Suggested Biomolecular Assignment
4000–3500	4000–3500	O-H, v	water
3350	3357	O-H, v	saccharides
2920	2920	CH_2_, CH_3_ν_as_	lipids (cutin and waxes), proteins, carbohydrates
2851	2850	CH_2_, ν_s_	lipids (cutin and waxes), proteins, carbohydrates
2442–2208	2442–2208	C-O-C, v	CO_2_
1730	1735	C=O, v	lipids (cutin and waxes)
1633	1629	C-O, v C-N, v	amide I band
1545		Amide II bands	proteins
	1528	Amide II bands	phenolic acids, flavonoids
14,401,374	14,401,376	O-H, v O-H, v	organic acid, flavonoids organic acid, flavonoids
1400		C-H, δ	saccharides
1321 1259	1314 1260	C-O, v C-O, v	lipids, flavonoid lipids, flavonoid
1147	1152	C-O, v CO-O-C, ν_as_	cholesterol ester, oligosaccharides, triacylglycerols
1047	1049	C-O, v	starch
930		C-O-C, skeletal	saccharides
817	813	C-H, δ_oop_	
	781	COO−, skeletal	saponins

v—stretching, ν_s_—symmetrical stretching, ν_as_—asymmetrical stretching, δ—bending, δ_oop_—bending out of the plane.

**Table 2 molecules-27-04640-t002:** Comparing the performance of RF, PLS-DA, and SVM models according to various normalization and spectral filter transformations between LJF and LF samples.

RF Model					
Pretreatment Methods	SENS	SPEC	ACC	MCC	AUC
No methods	0.9167	0.8250	0.8750	0.7481	0.9190
Vector (first)	0.9706	1	0.9844	0.9692	0.9710
Vector (second)	0.9583	1	0.9773	0.9554	0.9630
Min-max	0.9375	0.9750	0.9545	0.9097	0.9368
Area	0.9500	0.9375	0.9432	0.8859	0.9491
EWMA	0.9167	0.9000	0.9091	0.8167	0.9090
MSC	0.9500	0.9583	0.9545	0.9083	0.9310
RC	0.9500	0.9583	0.9545	0.9083	0.9430
S-G	0.9250	0.9167	0.9205	0.8401	0.9168
SNV	0.9500	0.9375	0.9432	0.8859	0.9291
airPLS	0.9750	0.9792	0.9773	0.9542	0.9690
PLS-DA Model					
Pretreatment methods	SENS	SPEC	ACC	MCC	AUC
No methods	0.9412	1	0.9687	0.9393	0.9229
Vector (first)	0.9750	0.9792	0.9773	0.9542	0.9710
Vector (second)	0.9500	0.9583	0.9545	0.9083	0.9630
Min-max	0.9333	0.9667	0.9687	0.9389	0.9218
Area	0.9118	0.9706	0.9531	0.9104	0.9091
EWMA	0.9412	1	0.9687	0.9393	0.9290
MSC	0.9706	0.9667	0.9687	0.9373	0.9610
RC	0.9750	0.9792	0.9773	0.9542	0.9430
S-G	0.9412	1	0.9687	0.9393	0.9268
SNV	0.9667	0.9706	0.9687	0.9373	0.9491
airPLS	0.9750	0.9792	0.9773	0.9542	0.9690
SVM Model					
Pretreatment methods	SENS	SPEC	ACC	MCC	AUC
No methods	0.9500	0.9792	0.9659	0.9314	0.9390
Vector (first)	0.9792	0.9981	0.9716	0.9724	0.9710
Vector (second)	0.9750	0.9792	0.9773	0.9542	0.9630
Min-max	0.9750	0.9792	0.9773	0.9542	0.9668
Area	0.7250	0.5208	0.6136	0.2490	0.1291
EWMA	0.9500	0.9792	0.9659	0.9314	0.9290
MSC	0.9250	0.9792	0.9545	0.9089	0.9010
RC	0.9500	0.9583	0.9545	0.9083	0.9230
S-G	0.9500	0.9792	0.9659	0.9314	0.9168
SNV	0.9750	0.9792	0.9773	0.9542	0.9491
airPLS	0.9500	0.9792	0.9659	0.9314	0.9390

RF: random forest; PLS-DA: partial least squares-linear discriminant analysis; SVM: support vector machine regression; EWMA: exponentially weighted moving average; MSC: multiplicative scatter correction; RC: row center; S-G: Savitzky–Golay; SNV: standard normal variate; airPLS: adaptive iteratively reweighted penalized least squares.

**Table 3 molecules-27-04640-t003:** The classification results and evaluation parameters between LJF and LF combined with RF, PLS-DA, and SVM models by vector normalization applied after the first differentiation.

Calibration Set	SENS	SPEC	ACC	MCC	AUC
RF	0.9706	1	0.9844	0.9692	0.9775
PLS-DA	0.9750	0.9792	0.9773	0.9542	0.9546
SVM	0.9792	0.9981	0.9716	0.9724	0.9668
Validation set	SENS	SPEC	ACC	MCC	AUC
RF	0.9706	0.9981	0.9744	0.9592	0.9425
PLS-DA	0.9250	0.9792	0.9545	0.9089	0.9006
SVM	0.9500	0.9792	0.9659	0.9314	0.9218

**Table 4 molecules-27-04640-t004:** The parameter screening in the RF model for variables is ranked by permutation accuracy importance.

n_tree_	SENS	SPEC	ACC	MCC	AUC
100	0.9750	0.9792	0.9773	0.9554	0.9390
200	0.9286	1	0.9583	0.9188	0.9010
300	0.9706	1	0.9844	0.9692	0.9775
500	0.9583	0.9750	0.9659	0.9316	0.9168
800	0.9583	0.9750	0.9659	0.9316	0.9168
1000	0.9286	1	0.9583	0.9188	0.9018
m_try_	SENS	SPEC	ACC	MCC	AUC
82	0.9583	1	0.9773	0.9554	0.9390
84	0.9583	0.9750	0.9659	0.9316	0.9168
86	0.9792	1	0.9886	0.9774	0.9875
88	0.9706	1	0.9844	0.9692	0.9775
90	0.9583	0.9750	0.9659	0.9316	0.9168
92	0.9583	1	0.9773	0.9554	0.9390
94	0.9792	0.9750	0.9773	0.9542	0.9390
96	0.9583	0.9750	0.9659	0.9316	0.9168

**Table 5 molecules-27-04640-t005:** List of permutation parameters of the random forest model obtained using variables selected by vector normalization applied after the first differentiation.

Normalization Method	SENS	SPEC	ACC	MCC	AUC
Vector (First)					
4000–600 cm^−1^ except for water vapor, carbon dioxide region	0.9750	0.9792	0.9773	0.9542	0.9425
2000–600 cm^−1^	0.9792	0.9750	0.9773	0.9542	0.9390
4000–2000 cm^−1^	0.9583	1	0.9773	0.9554	0.9390
4000–600 cm^−1^	0.9706	1	0.9844	0.9692	0.9775

**Table 6 molecules-27-04640-t006:** Various VIP cutoff values using 4000–600 cm^−1^ wavenumber areas for the comparison of LJF and LF.

VIP Cutoff	SENS	SPEC	ACC	MCC	AUC
0.05	0.9412	1	0.9688	0.9393	0.9168
0.01	0.9750	1	0.9886	0.9773	0.9775
0.015	0.9512	0.9867	0.9731	0.9465	0.9425
0.020	0.9000	0.9706	0.9375	0.8758	0.8625

## Data Availability

Not applicable.

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
