# Peer review of "Discrimination and Prediction of Lonicerae japonicae Flos and Lonicerae Flos and Their Related Prescriptions by Attenuated Total Reflectance Fourier Transform Infrared Spectroscopy Combined with Multivariate Statistical Analysis"

_molecules, 2022, doi:10.3390/molecules27144640_

Round 1

Reviewer 1 Report

This study is more important because analysis of medicinal plants by various characterizations can explore the importance of plants in medicine. The analysis is supported by both experimental and statistical methods. The work is having more interest to the scientific community.

Author Response

Dear reviewer,
   Thank you very much for your comments and professional advice. These opinions help to improve the academic rigor of our article. Based on your suggestion and request, we have made corrected modifications to the revised manuscript. We hope that our work can be improved again. 
Reviewer1#
This study is more important because analysis of medicinal plants by various characterizations can explore the importance of plants in medicine. The analysis is supported by both experimental and statistical methods. The work is having more interest to the scientific community.
The author’s answer: Thank you very much for your review and confirmation.
Thank you very much for your attention and time. Look forward to hearing from you.
Yours sincerely,
Yang-qiannanTang
July 9, 2022
Key Laboratory for Quality Evaluation of Bulk Herbs of Hunan Province,
Hunan University of Chinese Medicine,
Hunan 410208, P R. China
Tel:0731-88458230
E-mail: tangyangqiannan@163.com

Reviewer 2 Report

The study it is well planned, but a reduced number of samples have been subjected to the analysis. For statistical studies a large number of samples generates a relevant result. So if more samples from different crops (years)/or other regions could be considered, it will improve the relevance of the results.

The obtained model it is not completely explained.

Please address the comments in the text

Author Response

Dear reviewer,

   Thank you very much for your comments and professional advice. These opinions help to improve the academic rigor of our article. Based on your suggestion and request, we have made corrected modifications to the revised manuscript. We hope that our work can be improved again. Furthermore, we would like to show the details as follows.

Reviewer2#

  1. Please revise “ntree”!

The author’s answer: Thank you for your correction, and we have revised it.

  1. Please revise or explain the term “mtry”.

The author’s answer: Thank you for your correction, and we have revised it.

3.3339cm-1 in Table 1, this absorbtion band it is not visible in the spectra. It is as well located in the streathing of O-H bond, therefore it is not posible to distingwish it under the water O-H vibration!.

The author’s answer: Thank you very much for your correction. We have revised it to 3357cm-1. However, we totally agree with you. It is not possible to distinguish it under the water O-H vibration. And it is easy to cause confusion. In this study, we finally retained the description of this item. Because, research showed that LJF and LF do contain a lot of saccharides, whose peaks are located in the stretching of the O-H bond. Moreover, all the tested materials were fully dried, and the air background of the ATR-FTIR has been deducted. Therefore, we finally retained this item.

4.2920cm-1 in Table 1, and all C-h bonds in which the C is saturated will give a vibration in this region no matter if it is CH, CH2 or CH3 group.

The author’s answer: Thank you very much for your review. The absorption peak of the CH2 group of lipids is indeed around 2920 cm-1. However, after checking some books and references, we find that the absorption value of the CH group generally does not exceed 2900 cm-1, and the CH3 group may appear in this range.

We could check the following literature:

 (https://doi.org/10.1016/j.foodcont.2021.107879ï¼›

https://doi.org/10.1016/j.foodchem.2008.08.063ï¼›

https://doi.org/10.1016/j.saa.2018.09.052ï¼›

https://doi.org/10.1002/bit.27915)

Therefore, we add the CH3 group to the table.

5.The literature it is bit out of date, the most recent citation is from 2018...so 4 years old. Please try to check the recent developments in the field.

The author’s answer: Thank you for your advice. We update the literature on developments in the field of Lonicerae japonicae Flosand Lonicerae Flos and their Related Prescriptions.

Thank you very much for your attention and time. Look forward to hearing from you.

Yours sincerely,

Yang-qiannanTang

July 9, 2022

Key Laboratory for Quality Evaluation of Bulk Herbs of Hunan Province,

Hunan University of Chinese Medicine,

Hunan 410208, P R. China

Tel:0731-88458230

E-mail: tangyangqiannan@163.com

Reviewer 3 Report

The authors' work is highly interesting, well achieved, and well-structured, demonstrating that the authors are knowledgeable regarding their subject. It combines the use of efficient MIR spectra preprocess methods with advanced machine learning methods such as RF, SVM, and PLS -DA to discriminate between Lonicerae japonicae Flos (LJF) and  Lonicerae Flos (LF).

However, my main concern is the work's originality (see comments below), and I'm curious what value the authors have added in this study. Indeed, multiple researchers have already researched the discrimination of LJF and LF using classical and modern approaches (chemometrics), which contradicts the authors' statement in the introduction.

The paper could well be accepted for publication only if the authors correct and respond to the following comments:

Previous work has already been published addressing the topic of LJF and LF discrimination, including that of Jie Feng et al. (DOI: 10.1007/s11694-018-9834-0)  in which they used machine learning methods such as extreme learning machine and back-propagation neural network to identify the two plant kinds.

1-      I ask the authors to specify, particularly in the introduction, what their work's originality or added values are.

2-      In the introduction: Remove the last sentence: “The present study is the first to investigate a classification model that can identify LJF and LF and their related prescriptions by ATR-FTIR spectroscopy” .

3-      Please add the references of previous works related to the identification of LJF and LF :

Jie Feng et al. Journal of Food Measurement and Characterization 12(8), 10.1007/s11694-018-9834-0

Infrared Physics & Technology, Volume 104,https://doi.org/10.1016/j.infrared.2019.103139

Phytochem Anal. 2019, 10.1002/pca.2882

Molecules. 2019 Oct; 24(19): 3455. doi: 10.3390/molecules24193455

Journal of Molecular Structure, Volume 1124, 15 November 2016, Pages 110-116

4-      Since there are many abbreviations in the paper, I recommend including a list of abbreviations such as SNV, MSC, PLS-DA, SMV, SENS, SPEC, ACC, MCC, etc.

5-      In the introduction, line 93: replace "study" with "work or paper".

6-      Line 102, add (RF) after Random Forest analysis.

7-      Line 141: "For" with  f minuscule.

8-      Line 237: cm-1 not cm-1.

9-      Table 1: corrects in the 2nd line 3357cm-1 instead of 3339cm-1.

10-  Because the two MIR spectra in Figure 1 show that it is difficult to distinguish between FJL and FL, can it be possible to use NIR spectra to discriminate between FJL and FL ?

11-  Is there a possibility to render the data in table 2 and 3 into graphs to facilitate the reading and visualization of the classification results?

Author Response

Dear reviewer,

   Thank you very much for your comments and professional advice. These opinions help to improve the academic rigor of our article. Based on your suggestion and request, we have made corrected modifications to the revised manuscript. We hope that our work can be improved again. Furthermore, we would like to show the details as follows.

Reviewer3#

The authors' work is highly interesting, well achieved, and well-structured, demonstrating that the authors are knowledgeable regarding their subject. It combines the use of efficient MIR spectra preprocess methods with advanced machine learning methods such as RF, SVM, and PLS-DA to discriminate between Lonicerae japonicae Flos (LJF) and Lonicerae Flos (LF).

However, my main concern is the work's originality (see comments below), and I'm curious what value the authors have added in this study. Indeed, multiple researchers have already researched the discrimination of LJF and LF using classical and modern approaches (chemometrics), which contradicts the authors' statement in the introduction.

The paper could well be accepted for publication only if the authors correct and respond to the following comments:

Previous work has already been published addressing the topic of LJF and LF discrimination, including that of Jie Feng et al. (DOI: 10.1007/s11694-018-9834-0)  in which they used machine learning methods such as extreme learning machine and back-propagation neural network to identify the two plant kinds.

1-I ask the authors to specify, particularly in the introduction, what their work's originality or added values are.

The author’s answer: Thank you for your professional advice. There are a few similarities between this study and this study. Actually, they are different. Firstly, in terms of the object of study, Feng’s study (DOI: 10.1007/s11694-018-9834-0) is mainly concerned with the study of LJF and LF as tea Leaves. Tea belongs to the field of food. And more importantly, they did not study the ability of the model to discriminate when other tea leaves (or other natural plants) are involved. However, our study mainly focuses on the study of LJF and LF as Chinese medicinal herbs. Their related prescriptions (it contains a variety of Chinese herbal medicines except for LJF and LF) are also discussed. LJF and LF related pharmaceutical industry is a big business in China. The adulteration problem in the pharmaceutical field is far more important than that in the food field, and the harm is great. Secondly, in terms of the result of the study, Feng’s study just establishes a single valid model that can identify LJF and LF effectively and nondestructively and has the potential for the identification of true and false LJF Tea Leaves. However, our study establishes a model based on the LFJ and LJ (both of them are Herbal medicine). What is more important, we applied this model to their related prescriptions. Finally, we proved that this method has good accuracy and applicability. Thirdly, in terms of the test method, hyperspectral imaging is applied in Feng’s study, and we use ATR-FTIR. Generally speaking, ATR-FTIR is cheaper and easier to obtain. Therefore, our research has wider applicability.

2-In the introduction: Remove the last sentence: “The present study is the first to investigate a classification model that can identify LJF and LF and their related prescriptions by ATR-FTIR spectroscopy”.

The author’s answer: Thank you for your professional advice. I have removed the last sentence from the paper.

3-      Please add the references of previous works related to the identification of LJF and LF :

Jie Feng et al. Journal of Food Measurement and Characterization 12(8), 10.1007/s11694-018-9834-0

Infrared Physics & Technology, Volume 104,https://doi.org/10.1016/j.infrared.2019.103139

Phytochem Anal. 2019, 10.1002/pca.2882

Molecules. 2019 Oct; 24(19): 3455. doi: 10.3390/molecules24193455

Journal of Molecular Structure, Volume 1124, 15 November 2016, Pages 110-116

The author’s answer: Thank you very much for your precious advice. I have added literature on the identification and related development of Lonicerae japonicae Flos and Lonicerae Flos to the paper, as have the 3 references you provided.

4-      Since there are many abbreviations in the paper, I recommend including a list of abbreviations such as SNV, MSC, PLS-DA, SMV, SENS, SPEC, ACC, MCC, etc.

5-      In the introduction, line 93: replace "study" with "work or paper".

6-      Line 102, add (RF) after Random Forest analysis.

7-      Line 141: "For" with  f minuscule.

8-      Line 237: cm-1 not cm-1.

9-      Table 1: corrects in the 2nd line 3357cm-1 instead of 3339cm-1.

The author’s answer (4-9): Thanks for your patience in reviewing. Regarding the details you pointed out, we have corrected them in the corresponding section of the article and added a list of abbreviations at the end of the paper.

10- Because the two MIR spectra in Figure 1 show that it is difficult to distinguish between FJL and FL, can it be possible to use NIR spectra to discriminate between FJL and FL?

The author’s answer: Thanks for your precious suggestions. NIR spectra are one of the most rapidly developing high-tech analytical techniques in the last decade, which has the advantages of rapidity, simplicity, and efficiency. But there is still a big difference between NIR spectra and MIR spectra. First, the objects of the compound groups of both are different, the wave number of NIR spectra is in the range of 12820-3959 cm-1, while that of MIR spectra is in the range of 4000-400 cm-1. Second, The NIR spectra are mainly related to the vibrational ensemble and signal frequencies of the hydrogen-containing groups, and the absorption bands are relatively wide with a severe overlap in each absorption band. It cannot be analyzed qualitatively directly currently, and it can only be analyzed by building calibration models with chemometric methods. The fingerprint region (1330-400 cm-1) of MIR spectra can precisely identify the compounds and their functional groups contained in the samples, and the characteristic absorption band (4000-1330 cm-1) only shows the absorption peaks of groups with large folded masses and bonding constants, and the absorption peaks are small and easy to identify, and both the fingerprint region and the characteristic absorption band can be used for the qualitative and quantitative analysis of compounds.

In short, if we only look at the spectra, MIR has stronger specificity and excellent structural feature recognition ability, while the NIR spectra have no specific characteristics. Since LJF and LF belong to the same family of plants, the chemical structure, and composition of their parts are very similar, and NIR spectroscopy may be unable to make a clear distinction in this regard. However, when combined with the chemometrics method, NIR and MIR are both effective means of identification.

11- Is there a possibility to render the data in Tables 2 and 3 into graphs to facilitate the reading and visualization of the classification results?

The author’s answer: Thank you for your review. I agree with you that images are visually better than tables. But the tables in this paper are not really easy to convert, after all, there are too many parameters to optimize. If it is transformed into images, there will be dozens of images, which is a large number, and the images do not show the results of the specific classification parameters of the model. Therefore, we still present this part of the paper in the form of a table.

Thank you very much for your attention and time. Look forward to hearing from you.

Yours sincerely,

Yang-qiannanTang

July 9, 2022

Key Laboratory for Quality Evaluation of Bulk Herbs of Hunan Province,

Hunan University of Chinese Medicine,

Hunan 410208, P R. China

Tel:0731-88458230

E-mail: tangyangqiannan@163.com

Round 2

Reviewer 2 Report

The article was improved.

Reviewer 3 Report

The authors' responses and corrections are acceptable and may be considered.

1- However, in regards to point 11, I am not convinced by the authors' response. Indeed, because the goal of this paper is to develop classification models, a curve known as the "receiver operating characteristic curve" (ROC), in which the sensitivity (recal) is plotted as a function of (1-spécificity), is important. This graphic is required to visualize the effectiveness of classifiers developed with RT, SVM, and PLS-DA.

2- The abbreviations must be organized alphabetically.

3- The abbreviation table should be placed at the beginning of the article.